# Rule-Based Reference Updates after R1-Based Post Reinforcement Learning For Small Reasoning Language Models.

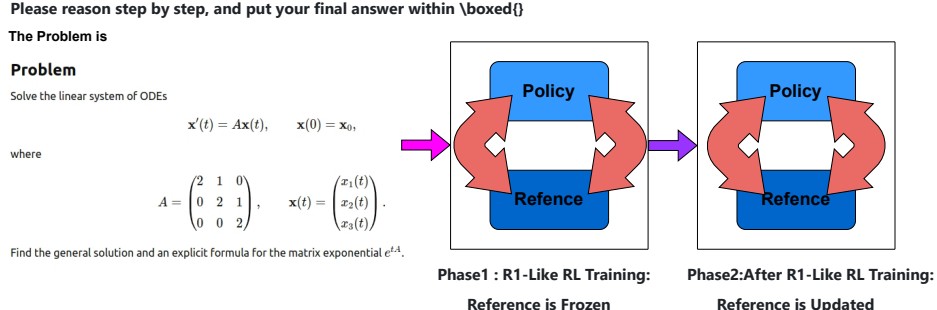

Figure 1: From Phase 1 : R1-Like Reinforcement Learning and its Curriculum Learning Variants To Phase 2: Reference Model Updates in Reinforcement Learning.

## Abstract

Inference scaling improves LLM reasoning, with reinforcement learning as a key driver. Although, post-training reinforcement learning and its curriculum learning variants offer significant benefits in enhancing the reasoning ability of large language models, we designate this process as Phase 1. Following this, we propose Phase 2: rule-based reference model updates in reinforcement learning after Phase 1 to explore the potential of reference model updates following R1-Like reinforcement learning. In details, we introduce a rule-based reference updates reinforcement learning approach continues to enhance the reasoning capabilities of small-sized large language models after current classical post-training reinforcement learning. In particular, a $1.5B$-parameter LLM achieves $60.2\%$ on AIME24, $48.2\%$ on AIME25 and $91.5\%$ on Math500 and $1.5\% - 4\%$ score improvement on AMC, Minera and Olympia. These results, enabled by the proposed rule-based reference model updates reinforcement learning algorithm, demonstrate math reasoning capabilities comparable to O1-mini/O3-mini—achievable within a typical school laboratory setting. In addition, we open-source both the dataset and model checkpoints to support future research in large-scale reinforcement learning for LLMs.

## 1 Introduction

Large language models (LLMs) like OpenAI's o-series Jaech et al. (2024); OpenAI (2024; 2025c;d), DeepSeek R1 Guo et al. (2025), Claude 3.7 Anthropic (2025), Grok3 XAI (2024), and Gemini 2.5 LLC (2025) excel in tasks such as math reasoning and code generation. These models leverage large-scale reinforcement learning (RL) to acquire advanced reasoning strategies—step-by-step analysis Wei et al. (2022), self-reflection Wang et al. (2023), and backtracking Ahmadian et al. (2024). While prior RL success has relied on large base models, enhancing reasoning in small models remains challenging. To address this, we explore the reference model updates after R1-Like reinforcement learning and its curriculum learning variants approaches Guo et al. (2025); Christiano et al. (2017);

Sutton & Barto (2018); Everitt et al. (2017; 2021); Weng (2024) for improving reasoning capabilities in smaller LLMs.

In this work, we introduce the rule-based reference model updates in GRPO algorithm, a post-training reinforcement learning approach tailored for small to medium-sized LLMs after the process of R1-Like reinforcement learning and its curriculum learning variants Figure 1. Our 1.5B-parameter models trained with this method outperform larger closed- and open-source reasoning models—including OpenAI's O1-mini and O1 OpenAI (2024); Jaech et al. (2024)—on key math reasoning benchmarks, demonstrating strong reasoning capabilities.

## 2 RELATED WORK

### 2.1 REASONING LARGE LANGUAGE MODELS

Reinforcement learning (RL) has been key to aligning LLMs with human preferences Christiano et al. (2017); Ouyang et al. (2022); Yuan et al. (2024a); Azar et al. (2024); Rafailov et al. (2023), while open-source efforts often rely on imitation learning for reasoning Yuan et al. (2024b); Yue et al. (2023); Guan et al. (2025). Recent models like OpenAI's O1 Jaech et al. (2024), DeepSeek R1 Guo et al. (2025), and Grok-3 XAI (2024) demonstrate the scalability of outcome-based RL. PRIME Cui et al. (2025) explores dense rewards, in contrast to the dominant outcome-only approaches Guo et al. (2025); Rafailov et al. (2023); Shao et al. (2024). Top models such as OpenAI's O-series Jaech et al. (2024); OpenAI (2024; 2025c;d), Claude 3.7 Anthropic (2025), and Gemini 2.5 LLC (2025) show strong math Guo et al. (2025); Jaech et al. (2024) and coding OpenAI (2025d); LLC (2025) reasoning. Long-COT models like O3 and DeepSeek-R1 OpenAI (2025c); Guo et al. (2025) benefit from RL with verifiable rewards (RLVR) Gandhi et al. (2025), avoiding costly MCTS-based data Hosseini et al. (2024); Yang et al. (2024), though they often overthink simple tasks Wang et al. (2024); Kumar et al. (2025). Efficiency-focused alternatives include latent-space optimization Hao et al. (2024); Geiping et al. (2025) and early-exit strategies Muennighoff et al. (2025); Fu et al. (2024); Zhang et al. (2024). But deep RL remains underexplored for small-scale LLMs (0.7B–1.5B) trained with limited data and resources. Besides, the updates of reference model in the reinforcement learning for enhancing small-scale LLM reasoning with limited math data is not well explored.

### 2.2 HIERARCHICAL REINFORCEMENT LEARNING AND CURRICULUM LEARNING

Hierarchical Reinforcement Learning (HRL) supports temporal abstraction and efficient exploration Sutton & Barto (2018); Nachum et al. (2018), using frameworks like options Sutton & Barto (2018); Bacon et al. (2017); Harutyunyan et al. (2018); Klissarov et al. (2017); Kaelbling (1993); Gao et al. (2024b); Dayan & Hinton (1993a); Salter et al. (2022) and goal-conditioned feudal models Dayan & Hinton (1993b); Vezhnevets et al. (2017). Techniques like transition relabeling Nachum et al. (2018); Levy et al. (2018) and leveraging demonstrations Rajeswaran et al. (2018); Nair et al. (2018); Hester et al. (2018); Shiarlis et al. (2018); Fox et al. (2017); Kipf et al. (2019); Zhang et al. (2020); Pertsch et al. (2020); Chane-Sane et al. (2021); Kreidieh et al. (2020); Singh et al. (2021), behavior priors Salter et al. (2022), and action primitives Dalal et al. (2021); Nasiriany et al. (2022) further boost learning. However, the reference model updates in Hierarchical Reinforcement Learning and not well studied for further increase the reasoning ability of large language models after R1-Like reinforcement learning and it's curriculum learning variants. Therefore, we start to explore the reference model updates after the phase of R1-Like reinforcement learning and its curriculum learning variants.

## 3 METHOD

Although R1-Like reinforcement learning and its curriculum learning variants offer significant benefits in enhancing the reasoning ability of large language models, we designate this process as Phase 1. Following this, we propose Phase 2: rule-based reference model updates in reinforcement learning after Phase 1 to explore the potential of reference model updates following R1-Like reinforcement learning Figure 2. The details of the two phases are outlined below.

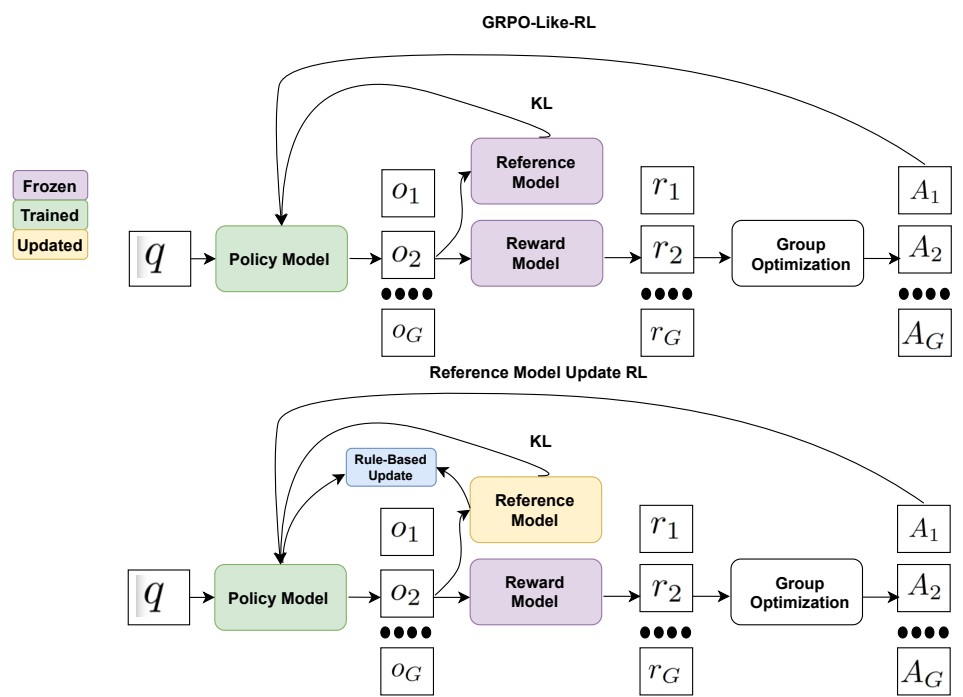

Figure 2: Description of Phase 1: GRPO Reinforcement learning and it's Variants with Frozen Reference Model. Phase 2: Reinforcement learning with Updated Reference Model in GRPO.

### 3.1 PHASE 1(PRELIMINARY): LLM REASONING VIA GRPO PLUS(GRPO+) AND VARIANTSYU ET AL. (2025)

#### 3.1.1 GROUP RELATIVE POLICY OPTIMIZATIONSHAO ET AL. (2024)

Compared to Proximal Policy Optimization (PPO) Schulman et al. (2017), Group-Relative Policy Optimization (GRPO) Shao et al. (2024) eliminates the value function and estimates the advantage in a group-relative manner.

For a specific question-answer pair $(q, a)$, the behavior policy $\pi_{\theta_{\text{old}}}$ samples a group of $G$ individual responses $\{o_i\}_{i=1}^G$. Then, the advantage of the $i$-th response is calculated by normalizing the group-level rewards $\{R_i\}_{i=1}^G$ as follows:

$$\hat{A}_{i,t} = \frac{r_i - \text{mean}(\{R_i\}_{i=1}^G)}{\text{std}(\{R_i\}_{i=1}^G)}. \tag{1}$$

Similar to PPO, GRPO adopts a clipped objective, together with a directly imposed KL penalty term:

$$\mathcal{J}_{\text{GRPO}}(\theta) = \mathbb{E}_{(q,a)\sim\mathcal{D},\{o_i\}_{i=1}^G\sim\pi_{\theta^{\text{old}}}(\cdot|q)}$$

$$\left[\frac{1}{G}\sum_{i=1}^G\frac{1}{|o_i|}\sum_{t=1}^{|o_i|}\left(\min\left(r_{i,t}(\theta)\hat{A}_{i,t},\right.\right.$$

$$\left.\left.\text{clip}\left(r_{i,t}(\theta), 1-\varepsilon, 1+\varepsilon\right)\hat{A}_{i,t}\right) - \beta D_{\text{KL}}(\pi_\theta\|\pi_{\text{ref}})\right)\right] \tag{2}$$

where

$$r_{i,t}(\theta) = \frac{\pi_\theta(o_{i,t} \mid q, o_{i,<t})}{\pi\_\theta_{\text{old}}(o_{i,t} \mid q, o_{i,<t})}. \tag{3}$$

It is also worth noting that GRPO Shao et al. (2024) computes the objective at the sample-level. To be exact, GRPO first calculates the mean loss within each generated sequence, before averaging the loss of different samples. Such difference may have an impact on the performance of the algorithm. where $\mu_R$ and $\sigma_R$ are the mean and standard deviation of the rewards in the group:

### 3.1.2 GROUP RELATIVE POLICY OPTIMIZATION PLUS(GRPO+)

The advanced Group Relative Policy Optimization algorithm Yu et al. (2025) is then developed. It samples a group of outputs $\{o_i\}_{i=1}^G$ for each question $q$ paired with the answer $a$, and optimizes the policy via the following objective:

$$\begin{aligned}
\mathcal{J}_{\text{GRPO}}(\theta) &= \mathbb{E}_{q \sim D_q, \{o_i\}_{i=1}^G \sim \pi_\theta(\cdot|q)} \\
&\left[ \frac{1}{\sum_{i=1}^G |o_i|} \sum_{i=1}^G \sum_{j=1}^{|o_i|} \min \left( \frac{\pi_\theta(o_i \mid q)}{\pi_{\theta_{\text{old}}}(o_i \mid q)} A_{i,j}, \right. \right. \\
&\left. \left. \text{clip} \left( \frac{\pi_\theta(o_i \mid q)}{\pi_{\theta_{\text{old}}}(o_i \mid q)}, 1 - \varepsilon_{\text{low}}, 1 + \varepsilon_{\text{high}} \right) A_{i,j} \right) \right]
\end{aligned} \tag{4}$$

where

$$A_{i,j} = \frac{r_i - \text{mean}(\{r_i\}_{i=1}^G)}{\text{std}(\{r_i\}_{i=1}^G)} \tag{5}$$

$$r_{i,t}(\theta) = \frac{\pi_\theta(o_{i,t} \mid q, o_{i,<t})}{\pi\_\theta_{\text{old}}(o_{i,t} \mid q, o_{i,<t})}. \tag{6}$$

Then, the key enhancements are represented as the following:

### 3.1.3 ENHANCEMENTS

Four kinds of enhancements to reasoning-focused reinforcement learning are applied. The KL penalty Kullback & Leibler (1951) is removed due to large divergence in long-CoT training. To improve exploration, *Clip-Higher* relaxes PPO's upper clipping bound Schulman et al. (2017). Dynamic sampling filters extreme-reward samples to maintain gradient diversity. Lastly, a token-level policy gradient KimiTeam et al. (2025) emphasizes longer responses for finer reward attribution.

### 3.1.4 CURRICULUM GRPO-LIKE REINFORCEMENT LEARNING.

Then, curriculum reinforcement learning is applied as an post-training strategy to further enhance the reasoning capabilities of large language models. The central idea is to structure the training process in a progressive manner, where the model is exposed to reasoning problems of gradually increasing difficulty rather than being overwhelmed by the full complexity of the task at once. At the early stages of training, the model begins with relatively simple reasoning instances that require shorter chains of thought, enabling it to establish a solid foundation of problem-solving skills and response structures. As training progresses, the complexity of the problems is systematically increased in a step-by-step curriculum, encouraging the model to develop more sophisticated reasoning strategies and adapt to diverse scenarios. This incremental exposure not only improves the robustness of reasoning but also optimizes the efficiency of the model's thinking process. By controlling both the difficulty of the reasoning tasks and the expected length of the responses within each sub-curriculum stage, the method guides the model to generate more concise and effective reasoning traces. As a result, the model learns not only to solve increasingly challenging problems but also to refine the quality, brevity, and accuracy of its outputs, ultimately leading to stronger reasoning performance with improved generalization across tasks.

## 3.2 PHASE 2: RULE-BASED REFERENCE MODEL UPDATES IN GROUP RELATIVE POLICY OPTIMIZATION(GRPO)

### 3.2.1 BATCH-SIZE-LEVEL TRAINING DATASET COLLECTION

In this phase of reinforcement learning, training dataset collection focuses on gathering data samples that can further strengthen the reasoning abilities of the policy model after the initial optimization stage, which includes R1-like reinforcement learning and curriculum-based variants. Unlike the earlier phase, where the goal is to build foundational reasoning skills and gradually increase task difficulty, this stage emphasizes the large-scale construction of high-quality reasoning trajectories to drive the next round of policy updates. The dataset is collected in batches, with each batch following the collection rules represented as the following:

$$D = threshold_2 < f_D(pass1@avg(rollout_n)) < threshold_1 \tag{7}$$

In detail, at each training step, we collect only a subset of each batch from the training dataset. For each sample, if the score of $rollout_n$ lies within the range $(threshold_2, threshold_1)$, the sample is retained in the batch. Samples with scores outside this range are discarded from the dataset. In the implementation, $threshold_2 = 0.2$ and $threshold_2 = 0.8$.

### 3.2.2 UPDATES IN STEP-WISE LEVEL

After the training dataset is collected at the batch-size level, we update the reference model in the GRPO training at the step-wise level. A rule-based update strategy is applied, in which the weights of the frozen reference model are replaced with the weights of the current actor model at specific training steps in Phase 2. In detail, at each training step $i$, we define an *updating score*, denoted as

$$\Delta\big(pass1@avg(rollout_n^i)\big), \tag{8}$$

which is calculated as the difference between $pass1@avg(rollout_n^i)$ and $pass1@avg(rollout_n^{i-1})$. Here, $pass1@avg(rollout_n^i)$ represents the $pass1@avg$ score of $rollout_n$ at training step $i$, with $n$ fixed at 8. The update timestep $update_{i*}$ is then chosen as the step $i^*$ where the updating score achieves its maximum value within the interval from step 0 to step $t_i$, where $t_i$ ¡ $t_{total}$, the total number of training steps in one epoch of Phase 2 reinforcement learning.

$$Update_{i*} = \max_{0 \leq i \leq t_i} \Delta\big(pass1@avg(rollout_n^i)\big)$$
$$\Delta\big(pass1@avg(rollout_n^i)\big) = pass1@avg(rollout_n^i) - pass1@avg(rollout_n^{i-1}) \tag{9}$$

In the implementation, to further enhance the stability of Phase 2 during reference model updating, we introduce an additional evaluation step before committing to any update. Specifically, at the timestep $Update_{i*}$, the actor model is evaluated on the AIME24 benchmark to obtain its performance score under the metric $pass1@avg16$. This additional evaluation serves as a safeguard to ensure that only meaningful improvements are transferred to the reference model. If, and only if, the actor model achieves a higher $pass1@avg16$ score on AIME24 compared to the current reference model, then the update is applied by replacing the weights of the reference model with those of the actor model. Otherwise, the reference model remains unchanged, thereby preventing unnecessary or unstable updates that could harm the learning process. By incorporating this rule-based evaluation mechanism, the system ensures that reference model updates reflect genuine progress in reasoning performance rather than transient fluctuations, ultimately leading to a more stable and reliable training process in Phase 2 reinforcement learning.

## 4 EXPERIMENT

To investigate the effectiveness of the proposed rule-based reference updating GRPO in Phase 2 on the reasoning capabilities of large language models (LLMs), we conduct a series of experiments. The experiments are designed to provide a comparative analysis against the state-of-the-art reasoning-oriented LLMs of different parameters, in particular, DeepSeek-R1-Distill-Qwen-1.5B Guo et al.

Table 1: Model Performance Comparison

| Model | MATH500 | AIME24 | AIME25 | AMC | Minerva | Olympia |
|---|---|---|---|---|---|---|
| **Close-Source** | | | | | | |
| O1-Preview | 85.5 | 44.6 | – | – | – | – |
| O1-Mini | 90.0 | 70.0 | – | – | – | – |
| O1 | 90.4 | 71.5 | – | – | – | – |
| Claude Sonnet | 82.2 | 23.3 | – | – | – | – |
| **Open-Source-Large** | | | | | | |
| *DeepSeek-R1* | 97.3 | 79.8 | – | – | – | |
| *Qwen3-235B* | 94.6 | 85.7 | – | – | – | |
| *Llama 4 Behemoth* | 95.0 | 78.0 | – | – | – | |
| *Kimi-1.5* | 96.2 | 77.5 | – | – | – | |
| *Qwen 2.5-72B* | 83.1 | 30.0 | – | – | – | |
| *Phi4-Reasoning-14B* | – | 81.3 | – | – | – | |
| *Llama 4 Maverick* | 18.0 | 64.0 | – | – | – | |
| **Open-Source-4B/7B** | | | | | | |
| *MIMO-7B* | 95.8 | 68.2 | – | – | – | – |
| *DeekSeek-Qwen-Distill-7B* | 92.8 | 55.5 | – | – | – | – |
| *Qwen3-4B* | - | 73.8 | – | – | – | – |
| **Open-Source-1.5B** | | | | | | |
| *DeepSeek-R1-QWEN-1.5B* | 82.8 | 28.8 | – | 62.9 | 26.5 | 43.3 |
| *STILL-3-1.5B-Preview* | 84.4 | 32.5 | – | 66.7 | 29.0 | 45.4 |
| *FastCuRL-1.5B-Preview* | 88.0 | 43.1 | – | 74.2 | 31.6 | 50.4 |
| *FastCuRL-1.5B-V2* | 89.3 | 47.5 | – | 77.0 | 32.8 | 53.3 |
| *Diff-Aware-1.5B-Preview* | 89.2 | 50.0 | 33.0 | 77.1 | 35.3 | 51.9 |
| *FastCuRL-1.5B-V2* | 89.3 | 47.5 | 30.0 | 77.0 | 34.7 | 54.5 |
| *FastCuRL-1.5B-V3* | 90.5 | 49.6 | 32.9 | 78.5 | 34.7 | 53.3 |
| *FastCuRL-1.5B-V3+(Ours)* | 91.5 | 53.6 | 39.2 | 79.6 | 35.2 | 57.9 |
| *OpenNemotron-1.5B* | 85.9 | 54.0 | 41.7 | 75.8 | 26.3 | -56.2 |
| *OpenNemotron-1.5B+(Ours)* | 88.0 | 60.2 | 48.2 | 78.2 | 29.2 | 59.1 |

(2025), STILL-3-1.5B-Preview RUC-AIBOX (2025), DeepScaler-1.5B-Preview Luo et al. (2025), FastCuRL-1.5B-Preview Chen et al. (2025) with 1.5B parameters, Qwen3-4B Yang et al. (2025), DeepSeek-R1-Distill-Qwen-7B Guo et al. (2025), MIMO-7B Xiaomi LLM-Core Team (2025) with middle-sized parameters, Llama 4 Maverick AI (2025c), Phi4-Reasoning-14B Abdin et al. (2025), Qwen 2.5-72B Team (2024), Kimi-1.5 Team (2025a), Llama 4 Behemoth AI (2025b), Qwen3-235B Team (2025b), DeepSeek-R1 Guo et al. (2025) with large-sized parameters, and closed-source reasoning models such as Claude 3.7 Sonnet (Standard) Anthropic (2025), O1, O1-Mini OpenAI (2024a), and O1-Preview OpenAI (2024b), enabling a thorough evaluation of the proposed methods.

## 4.1 EXPERIMENT SETUP AND EVALUATION

**Setup**  We use DeepSeek-R1-Distill-Qwen-1.5B Guo et al. (2025) as our base model, a 1.5B parameter distilled model. Training is conducted using the AdamW optimizer Loshchilov & Hutter (2019) with a constant learning rate of $1 \times 10^{-6}$. For roll-outs, we set the temperature to 0.6 and sample 16 responses per prompt, appending "Let's think step by step and output the final answer within \boxed{}." to each problem without a system prompt.

**Benchmarks and Dataset**  We evaluate our model across five math reasoning benchmarks: MATH500 Hendrycks et al. (2021), AIME2024 AI-MO (2024a), AMC2023 AI-MO (2024b), Minerva Lewkowycz et al. (2022), and OlympiadBench He et al. (2024b). For fair comparison, the

Table 2: Combined Model Rankings with Updated MATH500—AIME24—AIME25 Scores

| MATH-500 | | AIME24 | | AIME25 | |
|---|---|---|---|---|---|
| Model | Acc. | Model | Acc. | Model | Acc. |
| Gemini 2.5 Pro Exp | 95.2% | O3-Pro | 93.0% | o3-Mini | 86.5% |
| O3 | 94.6% | Gemini 2.5 Pro Exp | 92.0% | Gemini2.5-Pro-Exp | 85.8% |
| Qwen 3 (235B) | 94.6% | DeepSeek-R1-0528 | 91.4% | o3 | 85.3% |
| Grok 3 Mini Fast High Reasoning | 94.2% | O3 Mini | 86.5% | Grok3-Mini-Fast | 85.0% |
| O4 Mini | 94.2% | Gemini 2.5 Pro Exp | 85.8% | Qwen3-(235B) | 84.0% |
| DeepSeek R1 | 92.2% | O3 | 85.3% | o4-Mini | 83.7% |
| O3 Mini | 91.8% | Grok 3 Mini Fast High Reasoning | 85.0% | DeepSeek-R1 | 74.0% |
| Gemini 2.5 Flash Preview (Thinking) | 91.8% | Qwen 3 (235B) | 84.0% | o1 | 71.5% |
| Claude 3.7 Sonnet (Thinking) | 91.6% | O4 Mini | 83.7% | Grok3-Mini-Low-Reasoning | 70.6% |
| Gemini 2.5 Flash Preview | 91.6% | Qwen-3-30B-A3B | 65.8% | Phi-4 Reasoning | 65.8% |
| O1 | 90.4% | O1 | 71.5% | GPT-4.1 | 66.3% |
| **Ours-1.5B** | 91.5% | Grok 3 Mini Fast Low Reasoning | 70.6% | DeepSeek R1 | 74.0% |
| Grok 3 Beta | 89.8% | **Ours-1.5B** | 60.2% | Grok3 | 58.7% |
| DeepSeek V3(03/24/2025) | 88.6% | Grok 3 Beta | 58.7% | **Ours-1.5B** | 48.2% |
| Gemini 2.0 Flash(001) | 88.0% | DeepSeek V3 | 52.2% | DeepSeek-R1-Distill-70B | 46.7% |
| GPT4.1 Mini | 88.0% | GPT 4.1 mini | 49.4% | Gemini 2.0 Flash (001) | 29.8% |
| GPT4.1 | 87.2% | Claude 3.7 Sonnet (Thinking) | 44.6% | Claude 3.7 Sonnet (Non-thinking) | 22.5% |
| Mistreal Medium 3 | 87.0% | Mistreal Medium 3 | 42.3% | Gemini 1.5 Pro (002) | 18.7% |
| LLama4 Maveric | 85.2% | GPT4.1 | 39.8% | Gemini 1.5 Flash (002) | 17.3% |
| Gemini 2.0 Flash Think Exp | 84.6% | Gemini 2.0 Flash (001) | 29.8% | | |
| Gemini 1.5 Pro (002) | 82.8% | DeepSeek V3 | 27.5% | | |
| DeepSeek V3 | 80.4% | GPT4.1 nano | 27.3% | | |

Table 3: Ablation Studies

| Model | MATH500 | AIME24 | AIME25 | AMC | Minerva | Olympia |
|---|---|---|---|---|---|---|
| **Close-Source** | | | | | | |
| **Base1-FastCuRL-1.5B-V3** | 89.3 | 47.5 | 30.0 | 77.0 | 34.7 | 54.5 |
| **W/O** Data Collection | 89.3 | 47.5 | 30.0 | 77.0 | 34.7 | 54.5 |
| **W/O** Reference Update | 89.0 | 47.2 | 30.0 | 76.0 | 34.2 | 54.9 |
| **Ours** | 91.5 | 53.6 | 39.2 | 79.6 | 35.2 | 57.9 |
| **Base2-ReasoningNemotron-1.5B** | 85.9 | 54.0 | 41.7 | 75.8 | 26.3 | 56.2 |
| **W/O** Data Collection | 86.1 | 54.2 | 41.9 | 76.1 | 27.1 | 56.3 |
| **W/O** Reference Update | 85.8 | 54.1 | 41.9 | 76.0 | 26.9 | 56.8 |
| **Ours** | 88.0 | 60.2 | 48.2 | 78.2 | 29.2 | 59.1 |

training dataset is the same with the baseline models Wasi Uddin Ahmad (2025); Song et al. (2025). It includes problems of varied difficulty, comprising AIME of America (2024), AMC of America (2025), Omni-MATH Gao et al. (2024a), and Still RUC-AIBOX (2025). Similarly, the parameter setting such as the temperature, the $top_p$, the $top_k$, the max length of thinking tokens is settingas the same with the baseline models Wasi Uddin Ahmad (2025); Song et al. (2025).

**Evaluation Metric** We adopt PASS@1 as the evaluation metric. Using a temperature of 0.6 and top-$p = 1.0$, we generate $k = 16$ responses per question. PASS@1 is then computed as: PASS@1 $= \frac{1}{k} \sum_{i=1}^{k} p_i$.

## 4.2 MATH REASONING EXPERIMENTS

**Math Benchmarks** The proposed rule-based reference model updates in phase 2 is evaluated against top open- and closed-source models, including Gemini-2.5-Pro DeepMind (2025a), O3-Mini OpenAI (2025a), Grok-3-Mini (High) xAI (2025a), Qwen3-235B-A22B Team (2025c), and others. As shown in Table 3, our 1.5B model achieves strong performance across benchmarks: 56.3 Pass@1 on AIME24 Jia (2025), 90.5 on MATH500 HuggingFaceH4 (2025), 78.6 on AMC23 of America (2023), 34.7 on Minerva Dyer & Gur-Ari (2022), and 55.5 on OlympiadBench He et al. (2024a),

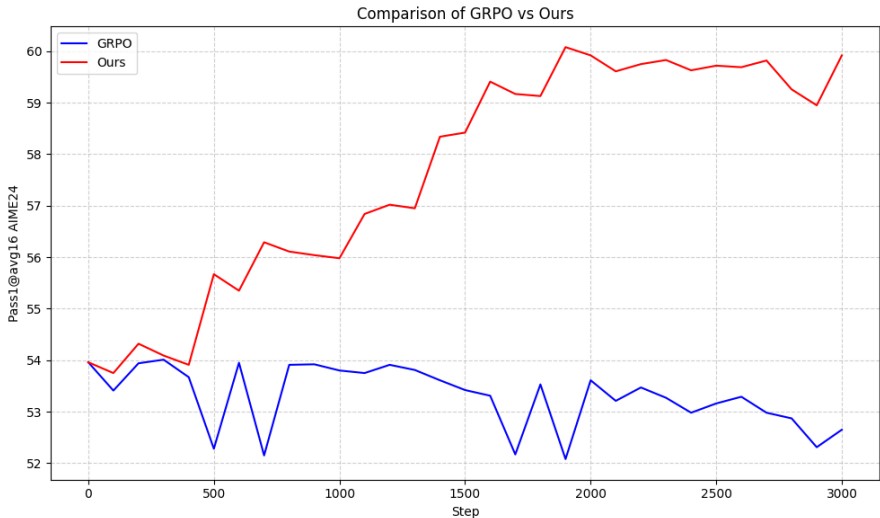

Figure 3: Comparison of Reference Frozen(GRPO) and Proposed Reference Updates(Oues) showing Pass1@avg16 AIME24 values over training steps.

demonstrating robust reasoning ability across diverse math tasks. For fair comparison, the length, the temperature and other hyper-parameters are set the same with the baseline1(FastCuRL-1.5B-V3 Song et al. (2025)) and baseline2(ReasoningNemotron-1.5B Wasi Uddin Ahmad (2025)) in the testing.

We evaluate our reasoning model against leading open- and closed-source models, including Gemini-2.5-Pro DeepMind (2025a), O3-Mini OpenAI (2025a), Grok-3-Mini (High) xAI (2025a), and Qwen3-235B Team (2025c). As shown in Table 3, our 1.5B model achieves strong results across diverse benchmarks: 60.2 on AIME24 Jia (2025), 91.5 on MATH500 HuggingFaceH4 (2025), 79.6 on AMC23 of America (2023), 39.2 on Minerva Dyer & Gur-Ari (2022), and 59.1 on Olympiad-Bench He et al. (2024a), demonstrating robust general mathematical reasoning.

For the Base1-FastCuRL-1.5B-V3 baseline, our approach achieves increases across all evaluation benchmarks. Notably, on MATH500, our model improves accuracy from 90.5% to 91.5%, which corresponds to an approximate 1.00% relative gain. On AIME24, the improvement is more significant, rising from 49.6% to 53.6%, a 4.00% increase. The gain on AIME25 is also notable, moving from 32.9% to 39.2%, which is a 7.00% relative improvement. Additional improvements can be seen on AMC (1.10%), Minerva (a slight decrease of 0.50%), and Olympia (4.50%). Compared with Base2-OpenReasoning-1.5B, our model shows even stronger gains. On MATH500, accuracy improves from 85.9% to 88.0%, representing a 3.10% improvement. On AIME24, the increase is from 54.0% to 60.2%, a 5.80% gain. Similarly, on AIME25, our model boosts performance from 41.7% to 48.2%, which corresponds to a 6.50% relative improvement. Further improvements are observed on AMC (2.40%) and Minerva (2.90%).

**Three Normal Math Leaderboards**   On a range of competitive benchmarks, our model demonstrates outstanding performance by ranking the 11th place on both the Math500 , 14th AIME24 leaderboards and 15th AIME25 leaderboards. In particular, on the Math500 HuggingFaceH4 (2025) benchmark, Ours-1.5B delivers exceptional results by outperforming several prominent models in the field. These include Grok 3 Beta xAI (2025b)(89.8%), DeepSeek V3 (03/24/2025) AI (2025a)(88.6%), Gemini 2.0 Flash (001) DeepMind (2025b)(88.0%). Similarly, on the AIME24 Jia (2025) benchmark, which evaluates problem-solving capabilities in a highly challenging math competition setting, Ours-$1.5B$ again achieves superior results. It surpasses DeepSeek V3 (03/24/2025) AI (2025a) (52.2%), GPT-4.1 Mini OpenAI (2025b) (49.4%). Besides, on the AIME25 Jia (2025) benchmark, which evaluates problem-solving capabilities in a highly challenging math competition setting, Ours-$1.5B$ again achieves superior results. It's score is almost the same with DeepSeek-

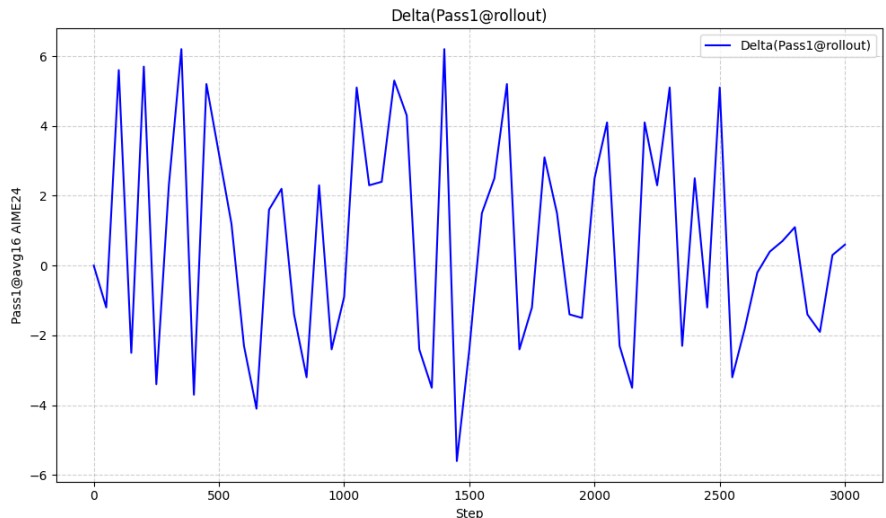

Figure 4: Details of showing $\Delta Pass1@avgrollout_{16}$ values over training steps for the phase 2.

R1-Distll-70B AI (2025a) and surpasses Gemini 2.0 Flash (001) AI (2025a) (29.8.2%), Claude 3.7 Sonnet (Non-thinking) 22.5% xAI (2025a).

**Ablation Studies** The ablation study in Table 3 investigates the contributions of individual components within two base models—Base1-FastCuRL-1.5B-V3 and Base2-Nemotron-Research-Reasoning-Qwen-1.5B across six benchmark datasets. For Base1, removing either Batch-Wise Data Collection or Step-Wise Reference Model Updation led to stagnation or a slight degradation in performance across tasks such as AIME24, AIME25, and Olympia, suggesting their necessity for reasoning enhancement. Notably, the full model outperforms all ablated variants, achieving 53.6% on AIME24 and 39.2% on AIME25. Similarly, in Base2, removing Batch-Wise Data Collection or Step-Wise Reference Model Collection leads to minor drops in performance.

**Key Visualization of Training Process** Visual details of the training process are shown in Figure3 and Figure4 for better comparison of the frozen reference model and updated reference model in the phase 2 and the values of updating score(in percentage%) in the phase 2. In details, in the phase 2 the pass1@avg16 score of frozen reference model(bule line) and updated reference model(red line) AIME24 is represented in Figure3. The updating score of the updated reference model is represented in Figure4.

## 5 DISCUSSION

We explore reinforcement learning of reference model updates to enhance the reasoning capabilities of large language models (LLMs) in mathematics. In Phase 2, our framework introduces rule-based updates to improve step-by-step problem-solving. We evaluate this approach on a small-scale math dataset using a 1.5B-parameter model.

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

## A APPENDIX

You may include other additional sections here.