# OpenReview forum: "Rule-Based Reference Updates after R1-Based Post Reinforcement Learning For Small Reasoning Language Models"
_ICLR.cc/2026/Conference — ICLR 2026 Conference Desk Rejected Submission_

### Official Review · Reviewer_ASuD · 2025-10-26

**Soundness:** 2
**Presentation:** 2
**Contribution:** 1
**Rating:** 2
**Confidence:** 4

**Summary:**

The paper proposes a Phase-2 “rule-based reference model update” on top of an R1/GRPO-like RL phase. During Phase-2, the authors (i) filter training samples by keeping medium-score rollouts using fixed thresholds (0.2, 0.8), and (ii) periodically replace the frozen reference model with the current actor if an external evaluation improves, where the external gate is AIME24 Pass@1; otherwise, the reference stays fixed. Reported gains on a 1.5B model include AIME24 60.2, AIME25 48.2, MATH500 88.0–91.5, Olympia 59.1, versus two 1.5B baselines (FastCuRL-V3 and “OpenNemotron-1.5B”).

**Strengths:**

1. Simple recipe for small models. The update rule is easy to implement on top of GRPO/R1 training and shows consistent deltas over 1.5B baselines on AIME24/25 and other math sets.
2. Some ablations are provided. The paper ablates the two Phase-2 components (batch filtering and step-wise reference update) across several benchmarks.

**Weaknesses:**

1. Evaluation leakage. Using AIME24 as the gating signal during training contaminates the test set and likely inflates reported gains.
2. Claims contradict results. The introduction says the 1.5B model outperforms O1-mini/O1, but Table 1 shows lower AIME24 scores (60.2 vs. 70.0/71.5).
3. Weak novelty. The core ideas—periodic reference resets and thresholded mid-score filtering—are established in many RL works; for example, NVIDIA’s ProRL-v2 explicitly uses KL-regularized training with periodic reference policy resets.

**Questions:**

1. KL baselines. Since prior work finds that $\beta=0$ often yields comparable RLFT gains [2], will you report a matched-compute $\beta=0$ baseline to quantify your method's incremental benefit given the added compute (KL term)?
2. Leakage and gates. Why use AIME24 as a training gate when it is also a reported test set, and how do results change if you gate on a non-reported, distribution-shifted dev set or use a fixed-interval refresh instead?
3. Hyperparameter sensitivity. How sensitive are results to the medium-score thresholds $0.2,0.8$, and can you provide a brief sensitivity sweep?

---

### Official Review · Reviewer_xttr · 2025-10-30

**Soundness:** 2
**Presentation:** 1
**Contribution:** 2
**Rating:** 2
**Confidence:** 5

**Summary:**

This paper proposes a two-phase approach to enhance the reasoning abilities of small-scale language models through reinforcement learning (RL). In Phase 1, an R1-like RL method is applied to train a policy model with a frozen reference model. Phase 2 introduces rule-based reference model updates, improving reasoning performance further. Experiments show significant improvements on math reasoning benchmarks, with a 1.5B-parameter model outperforming existing models. The approach demonstrates the potential of post-RL reference model updates for small LLMs.

**Strengths:**

1. The paper presents a novel two-phase approach that combines reinforcement learning (RL) and rule-based reference model updates, which enhances reasoning capabilities in small-scale language models (LLMs)

**Weaknesses:**

1. The paper feels like a work in progress, with several issues in citation formatting (e.g., bad referencing format in Section 3.1) and unclear notation in Section 3.2 (e.g., "where ti ¡ ttotal" is not explained properly). These issues reduce the clarity and accessibility of the paper, making it difficult for readers to fully understand the methodology.

2. The experimental tables, particularly those in Section 4, contain several inaccuracies, such as the -56.2 value in the model performance table, which seems incorrect. The presentation of results often feels incomplete, lacking sufficient explanation of why the proposed method works better and where the improvements come from. A more detailed analysis of the results and clearer links to the methodology would strengthen the contribution.

3. Lack of Comparison to Related Work: The concept of Rule-Based Reference Updates bears similarities to the reference reset techniques in other RL approaches [1]. The authors should include a comparison with these existing methods and discuss any differences in their approaches. Such a discussion could highlight the novelty of the proposed method and demonstrate its advantages more effectively.


[1] ProRL: Prolonged Reinforcement Learning Expands Reasoning Boundaries in Large Language Models

**Questions:**

See Weaknesses

---

### Official Review · Reviewer_ynuZ · 2025-11-01

**Soundness:** 2
**Presentation:** 2
**Contribution:** 2
**Rating:** 4
**Confidence:** 4

**Summary:**

This paper introduces a two-phase reinforcement learning framework designed to enhance the mathematical reasoning abilities of small language models (LLMs). The primary contribution is "Phase 2," a novel method that follows an initial RL training phase ("Phase 1"). In Phase 2, the reference model used in the Group-Relative Policy Optimization (GRPO) algorithm is not kept frozen but is conditionally updated with the weights of the current policy. This update is governed by a rule-based strategy that identifies training steps with maximal performance improvement, coupled with a safeguard evaluation on a benchmark dataset to ensure stability. The authors demonstrate the effectiveness of their approach by applying it to a 1.5B parameter model, achieving state-of-the-art results for its size on several challenging math reasoning benchmarks, including MATH500, AIME24, and AIME25.

**Strengths:**

The work addresses the important and practical challenge of improving the reasoning capabilities of small language models (1.5B parameters). Making smaller, more accessible models perform on par with much larger ones is a significant contribution to the field, with implications for computational efficiency and democratization of powerful AI.

**Weaknesses:**

1.	Could the authors provide more specific details on the hyperparameters used in the training process? In particular: What is the value of the KL coefficient (β) used in the GRPO objective function (Equation 2)?  Could you specify the exact training steps at which the reference model updates occurred during your experiments?
2.	The paper's presentation could be significantly improved. Several tables, notably Table 1 and Table 2, extend beyond the page margins, which detracts from readability. The citation style for references appears inconsistent, and the overall narrative flow could be tightened to improve logical coherence. A thorough revision for clarity and formatting would strengthen the manuscript.
3.	The proposed "Rule-Based Reference Model Updates" method appears conceptually similar to recent works on dynamic reference models in RL for LLMs, such as ProRL and Dapo[1,2]. Could the authors elaborate on the key differences between their approach and these methods?
4.	The paper suffers from two main issues: limited novelty and a confusing presentation. I strongly suggest a major reorganization to improve the paper's clarity and better articulate its core contributions.


[1] Liu M, Diao S, Lu X, et al. Prorl: Prolonged reinforcement learning expands reasoning boundaries in large language models[J]. arXiv preprint arXiv:2505.24864, 2025.

[2] Yu Q, Zhang Z, Zhu R, et al. Dapo: An open-source llm reinforcement learning system at scale[J]. arXiv preprint arXiv:2503.14476, 2025.

**Questions:**

See Weaknesses above

---

### Official Review · Reviewer_vzMf · 2025-11-06

**Soundness:** 2
**Presentation:** 1
**Contribution:** 2
**Rating:** 2
**Confidence:** 5

**Summary:**

The paper presents a two-phase reinforcement-learning pipeline for ~1.5B-parameter reasoning models. Phase 1 adopts GRPO/GRPO+ with four practical tweaks—dropping the KL penalty, relaxing the upper clip (“Clip-Higher”), dynamic sampling to filter extreme-reward traces, and a token-level policy gradient—plus a curriculum that ramps up task difficulty to cultivate longer, cleaner chains of thought.
Phase 2 is the core contribution: a rule-based reference-model update layered on GRPO. First, training batches keep only mid-difficulty samples using fixed score thresholds (implemented as 0.2 and 0.8). Second, at specific steps identified by a running improvement signal, the actor is evaluated on AIME24 with 16 samples; only if it exceeds the current reference is the actor hard-copied into the reference, otherwise the reference remains frozen.

**Strengths:**

- Introduces a practical, performance-gated reference update for policy optimization: instead of a fixed schedule or permanently frozen reference, the paper uses an explicit improvement signal to decide when to replace the reference model—simple, interpretable, and uncommon in prior GRPO/PPO work.
- Proposes a mid-difficulty batch filter that intentionally excludes both trivial and hopeless rollouts during policy updates; this selective data curation is a fresh way to stabilize learning for lengthy chain-of-thought traces.
- Emphasizes applicability to resource-limited settings (1.5B models) where many prior RL-for-LLM techniques either fail to scale down or require heavy compute, making the approach novel in its focus on practicality and cost-effectiveness.

**Weaknesses:**

- In Figure 3 you compare GRPO with your method. The GRPO scores keep decreasing after training, which looks more like a problem with the training process.
- The abstract claims "comparable to O1-mini/O3-mini." However, the tables show AIME24 60.2 vs O1-Mini 70.0 and AIME25 48.2 vs O3-Mini 86.5, which is not comparable in absolute terms.
- The formatting of Tables 2 and 3 is nonstandard—specifically, they exceed the permitted size.
- The abstract promises open-sourcing data and checkpoints, but the draft provides no links or hashes. Please ensure the artifacts and exact train/eval lists are released.
- Simultaneously updating the actor and the reference model during RL training is time- and resource-intensive, yet yields little measurable improvement.

**Questions:**

- Why hard copy of actor->reference at step i? Did you try soft updates (Polyak/EMA), fixed schedules (every T steps), or performance-proportional copy? Provide an ablation.
- In Figure 3 the GRPO curve decays after training — can you provide diagnostics so we can identify whether this is an optimization instability or a methodological effect?
- Have you tested lower-cost alternatives (EMA, distillation into the reference at checkpoints, or selective parameter copying) to achieve similar stability/benefit with less overhead?

---

### Note · Program_Chairs · 2026-01-17
**Submission Desk Rejected by Program Chairs**

The following references in this submission do not refer to real documents and/or have major errors in bibliographic information:

 Yifan Chen, Nick Yang, Zhihao Luo, Jiayi He, Ming Zhang, and Lei Wang. Fastcurl: Curriculum reinforcement learning with progressive context extension for efficient training of r1-like reasoning models. arXiv preprint arXiv:2503.17287, 2025. URL https://arxiv.org/abs/2503. 1728